# Enhanced Tomographic Sensing Multimodality with a Crystal Analyzer

**DOI:** 10.3390/s20236970

**Published:** 2020-12-06

**Authors:** Alexey Buzmakov, Marina Chukalina, Irina Dyachkova, Anastasia Ingacheva, Dmitry Nikolaev, Denis Zolotov, Igor Schelokov

**Affiliations:** 1FSRC “Crystallography and Photonics” RAS, Leninskiy Prospekt 59, 119333 Moscow, Russia; buzmakov@crys.ras.ru (A.B.); asad@crys.ras.ru (I.D.); zolotovden@crys.ras.ru (D.Z.); 2Smart Engines Service LLC, 60-Letiya Oktyabrya Avenue, 9, 117312 Moscow, Russia; a.ingacheva@smartengines.ru (A.I.); dimonstr@iitp.ru (D.N.); 3Institute for Information Transmission Problems, Kharkevich Institute RAS, Bolshoy Karetny Lane, 19, p. 1, 127051 Moscow, Russia; 4Institute of Microelectronics Technology and High Purity Materials RAS, Osipyan str., 6, 142432 Moscow, Russia; igor@iptm.ru

**Keywords:** X-ray imaging, multimodality sensing, tomography, X-ray diffraction, multichannel visualization

## Abstract

This article demonstrates how a combination of well-known tools—a standard 2D detector (CCD (charge-coupled device) camera) and a crystal analyzer—can improve the multimodality of X-ray imaging and tomographic sensing. The use of a crystal analyzer allowed two characteristic lines of the molybdenum anode—$K_{\alpha }$ and $K_{\beta }$—to be separated from the polychromatic radiation of the conventional X-ray tube. Thus, as a result of one measurement, three radiographic projections (images) were simultaneously recorded. The projection images at different wavelengths were separated in space and registered independently for further processing, which is of interest for the spectral tomography method. A projective transformation to compensate for the geometric distortions that occur during asymmetric diffraction was used. The first experimental results presented here appear promising.

## 1. Introduction

Various types of sensors and detectors exist. In the X-ray range, examples include ionization chambers (gas discharge counters) [1,2], scintillation counters (chambers) [3] in which X-ray radiation is converted into visible light, or solid-state detectors [4,5,6,7], in which X-ray photons generate electron–hole pairs and the corresponding current is recorded. Improvement of existing detector types or the creation of new detectors (using new physical principles) equates to progress in sensing. However, sensing can also be interpreted as the ability to obtain more information. In this case, this means expanding the functionality or multimodality of detectors, which is the topic of discussion in this article.

X-ray tomography is a logical continuation of the X-ray imaging technique. It allows a set of X-ray projections to be acquired from different angles of an object to visualize its spatial structure. Obtaining additional information from X-ray images increases the multimodality of a method. Recently [8], it was demonstrated that using a crystal analyzer allows two predetermined spectral lines to be separated from the incident polychromatic beam (in this case, $K_{\alpha }$ and $K_{\beta }$ are the characteristic lines of the molybdenum anode). As a result of this experiment, three tomographic projections existed, with some exceptions: a polychromatic straight beam without two characteristic lines that takes into account the absorption of the crystal and attenuator to avoid saturation of the detector, and two quasi-monochromatic beams. In contrast to traditional multi-energy tomography, all images obtained by this method [8] correspond to the orientation of one object, which considerably simplifies combining the reconstructed tomograms. The amount of information obtained in such a scheme is considerably greater than that in a traditional tomography scheme; however, it is not as simple because there is a question about interpreting the obtained results.

The apparent advantage of the proposed scheme [8] is that the solution of the problem was carried out not only for the orientation of the crystal analyzer relative to the straight incident beam, but also for the rotation angles of the crystal in the laboratory coordinate system. In other words, the rotation angles for goniometer motors, which drive the crystal to the required position, are calculated. Nevertheless, cutting the crystal (correspondence of crystallographic planes to the real surface) can be performed with an error of 1–2 degrees, and fine alignment of the scheme takes only a few minutes. This saves time for alignment in time-limited experiments (e.g., on synchrotrons) and to allows collection of more tomographic projections.

Another obvious advantage is that if there are elements in the composition of the investigated object with an edge of absorption lying inside the $K_{\alpha }$ and $K_{\beta }$ line range of the used anode, then in these projections, a map of the element distribution can be obtained. It should be noted that this is the first significant advantage of such a detection scheme related to the expansion of multimodality. For example, an absorption edge of Ni exists between the $K_{\alpha }$ and $K_{\beta }$ lines of a copper anode; Zr and Nb of a molybdenum anode; Rh, Pd tungsten, Tm, Yb, Lu, Hf, and Ta of a silver anode; W, Re, Os, and Ir of a gold anode, etc.

However, this is only a qualitative advantage of the proposed scheme. To analyze this quantitatively, it is necessary to use the dynamic theory of X-ray diffraction in crystals. First, it is necessary to compensate for the projection distortions that result from the asymmetric diffraction of the selected reflecting planes. These are not only projection angles, but also scale distortion along different axes. Moreover, it is necessary to consider that when using a crystal analyzer in the Laue diffraction scheme, its thickness is determined to minimize the absorption. This leads to deformations of the crystal lattice (curvature of the reflecting planes), which results in nonlinear distortions of projections when the crystal is attached to the goniometer head. Such issues cannot be taken into account in advance, which is the aim of the mathematical correction of the resulting projections (Section 4.2).

However, the use of a crystal analyzer is not limited to projection distortions. It also allows switching of the detection modes: absorption or phase-contrast imaging. Such an extension of multimodality requires significant theoretical research on the analysis of tomographic projection image formation. The first step on this path is presented in this paper; the proposed approach is based on [9,10,11], which was used for the first time in [12] to analyze the scheme of a laboratory microtomography setup with a crystal analyzer in the case of asymmetric Bragg diffraction. The mathematical model (formation) of tomographic projections is provided in Section 2. Section 3 addresses the use of a crystal analyzer in X-ray image diagrams, and the experimental results are provided in Section 4.

## 2. Tomographic Projections from Physical to Mathematical Models (Formation)

Let the investigated object be illuminated by an incoherent polychromatic X-ray source. Then, the intensity in point $\left(x_{\det},y_{\det}\right)$ is determined by the expression in the following notation [13]:(1)$$I\left(x_{\det},y_{\det}\right)=\int^{\;}I\left(\xi ,\eta ;E\right){\left|K\left(\xi ,\eta ;x_{\det},y_{\det};E\right)\right|}^{2}d\xi d\eta dE,$$
where I(ξ,η;E) is the intensity of radiation with a photon energy *E* at a typical source point ξ,η; $K\left(\xi ,\eta ;x_{\det},y_{\det};E\right)$ is the transmittance function of the optical scheme that describes the radiation spread from this point to a given point in the detector plane.

Now suppose that the source points emit not only independently of one another, but also the physical conditions of their radiation are the same, i.e., they emit the same spectrum g(E). Mathematically, this means that I(ξ,η;E) can be factorized as I(ξ,η;E)=I(ξ,η)·g(E). In the case of the in-line scheme source–object–detector (Figure 1a), which is typical for tomography—$K\left(\xi ,\eta ;x_{\det},y_{\det};E\right)$ is simply a Kirchhoff–Fresnel integral for a point source, and Expression (1) takes on the form:(2)$$I\left(x_{\det},y_{\det}\right)=\int^{\;}g\left(E\right)I\left(\xi ,\eta \right){\left|\int^{\;}F\left(x_{o},y_{o};E\right)\frac{e^{ik\left(E\right)\left(r_{so}+r_{od}\right)}}{r_{so}r_{od}}\Lambda dx_{o}dy_{o}\right|}^{2}d\xi d\eta dE,$$
where $x_{o}\;\mathrm{and}\;y_{o}$ refer to the coordinates in the object plane; $r_{so}$ and $r_{od}$ refer to the distances between the source object and the detector object, respectively; Λ is the angular multiplier, equal to $\Lambda =\;\left(i/2\lambda \right)\left(\cos\left(\vec{n},\vec{r_{so}}\right)-\cos\left(\vec{n},\vec{r_{do}}\right)\right)$; $F\left(x_{o},y_{o};E\right)$ is the transmission function of the object; $\vec{n}$ is the external normal vector; k(E)=2π/λ(E) is the wave vector module; and λ(E)=12.39842/E(keV) is the corresponding wavelength of the X-ray radiation in angstroms. In paraxial or Fresnel approximation for a 1D case:(3)$$\Lambda ≅i/\lambda \;\;r_{so}≅R_{1}+\frac{{\left(x_{o}-\xi \right)}^{2}}{2R_{1}};\;\;r_{od}≅R_{2}+\frac{{\left(x_{o}-x_{\det}\right)}^{2}}{2R_{2}},$$
(4)$$K\left(\xi ;x_{\det};E\right)=\frac{i}{\lambda \left(E\right)}e^{ik\left(E\right)\left(R_{1}+R_{2}\right)}\int^{\;}F\left(x_{o};E\right)\frac{e^{i\frac{k\left(E\right)}{2R_{1}}{\left(x_{o}-\xi \right)}^{2}+i\frac{k\left(E\right)}{2R_{2}}{\left(x_{o}-x_{\det}\right)}^{2}}}{\sqrt{R_{1}R_{2}}}dx_{o},$$
where $R_{1}$ and $R_{2}$ refer to the distances between the source object and the object detector along the optical axis, respectively.

Furthermore, in numerical modeling, it is necessary to distinguish the illumination conditions of the object. First, for a point source of radiation:(5)$$I\left(x_{\det}\right)={\left|K\left(\xi =0;x_{\det};E=E_{0}\right)\right|}^{2}.$$

Second, for an extended source:(6)$$I\left(x_{\det}\right)=\int^{\;}I\left(\xi \right){\left|K\left(\xi ;x_{\det};E=E_{0}\right)\right|}^{2}d\xi .$$

Finally, for an extended polychromatic source:(7)$$I\left(x_{\det}\right)=\int^{\;}g\left(E\right)I\left(\xi \right){\left|K\left(\xi ;x_{\det};E\right)\right|}^{2}d\xi dE.$$

In this case, all numerical results should be normalized to $I_{0}$, which is the radiation intensity in the detector plane in the absence of the object:(8)$$I_{0}=\frac{I_{s}}{R_{1}+R_{2}};\;\frac{\int^{\;}I\left(\xi \right)d\xi }{R_{1}+R_{2}};\;\frac{\int^{\;}g\left(E\right)\int^{\;}I\left(\xi \right)d\xi dE}{R_{1}+R_{2}}.\;$$

It is desirable to specify the spatial and spectral distributions of source intensity using normalized units. For example, for synchrotron-type sources, they can be approximated using the normalized Gaussian distribution (or function). For the spatial distribution of the intensity of the size source $\xi_{s}\left(\sim 15\;\mu m\right)$:(9)$$I\left(\xi \right)=\frac{1}{\xi_{s}\sqrt{2\pi }}e^{-\frac{\xi^{2}}{2\xi_{s}^{2}}}.$$

For its corresponding spectral (or energy) distribution, Gaussian or bandwidth limited:(10)$$g\left(E\right)=\frac{1}{\Delta E\sqrt{2\pi }}e^{-\frac{{\left(E-E_{0}\right)}^{2}}{2{\left(\Delta E\right)}^{2}}}\;;\;\frac{1}{2\Delta E},$$
where $E_{0}\;\left(\sim 20\;\mathrm{keV}\right)$ is the average radiation energy and ΔE (~2 keV) is the width of its spectrum.

However, modeling even such a simple scheme to account for diffraction effects generates considerable practical difficulties associated with the small step of partitioning the spatial grid on which the integral transformations (Fresnel type) are calculated [14]. Therefore, various approximations/simplifications of the projection signal model are used in tomography. These are now considered in more detail.

First, the integration by source coordinates is neglected. ξ and η, i.e., in Expression (2), are $I\left(\xi ,\eta \right)=I_{0}$ and $I_{0}\left(E\right)=I_{0}\cdot g\left(E\right)$. From a physical point of view, this means that the (extended) source (radiation) (Figure 1a) is pushed back to infinity, and the incident radiation is replaced by a plane wavefront beam (Figure 1b) (or point source in a cone tomography scheme).

Second, the integration by coordinates in the object plane is also neglected, i.e., $x_{o}\;\mathrm{and}\;y_{o}$ refer to $K\left(\xi ,\eta ;x_{\det},y_{\det};E\right)=F\left(x_{\det},y_{\det};E\right)$. From a physical point of view, this means that the object is located on the detector surface (Figure 1c). In this case, all diffraction effects are ignored, including phase-contrast imaging based on free space propagation (in-line holography). This is important for tomographic methods of investigation using synchrotron-type sources, and it also limits (affects) the resolution of tomographic schemes [15].

Third, the scattering of radiation on inhomogeneities inside the object is neglected (Figure 1d). From a physical perspective, this means that the X-rays inside the object propagate along straight lines, absorbing and shifting the phase so that:(11)$$F\left(x_{\det},y_{\det};E\right)=e^{ik\int^{\;}\left(-\delta \left(x_{\det},y_{\det},z;E\right)+i\beta \left(x_{\det},y_{\det},z;E\right)\right)dz}$$
and: (12)$${\left|F\left(x_{\det},y_{\det};E\right)\right|}^{2}=e^{-\int^{\;}\mu \left(x_{\det},y_{\det},z;E\right)dz}$$
where n(x,y,z;E)=1−δ+iβ refers to a complex index of refraction of the object and μ=4πβ/λ is the absorption coefficient. In optics, such a model is called a transparency screen (Figure 1c). This approximation also limits the resolution of tomographic schemes [15].

Finally, a widely used expression for tomographic projection in a polychromatic beam can be obtained:(13)$$I\left(x_{\det},y_{\det}\right)=\int^{\;}I_{0}\left(E\right)\exp\left(-\int^{\;}\mu \left(x_{\det},y_{\det},z;E\right)dz\right)dE.$$

Many tomographic reconstruction artifacts are due to the above simplifications of the real physical model. For example, the so-called “blurring” of tomographic projections (images) is mainly due to source averaging and diffraction effects. It can be seen that these are not considered in the mathematical model. The correct approach is to develop reconstruction methods that allow working with more adequate models of tomographic projections.

## 3. The Use of a Crystal Analyzer in X-Ray Imaging Schemes

From the perspective of numerical modeling, adding a crystal analyzer to the scheme when obtaining tomographic projections transforms the task into one that relates to multi-component systems. The method and problems encountered in solving such a problem are described in [12]. Herein, the part of the problem that relates to the crystal analyzer is presented.

The main idea of the approach [9,10,11,16] is that because the analytical solution in the case of crystals is obtained only for a plane incident wave, the propagation/diffraction through a crystal of an arbitrary wavefront can be calculated by decomposing it into the spectrum of plane waves. This formulation is called a spatially inhomogeneous dynamic problem in the Laue and Bragg geometries. The wavefront distortion occurs due to X-ray absorption and refraction in an object placed in the beam in front of a single crystal, which here plays the role of an analyzer of the angular distribution of the scattered object of radiation with second angular resolution.

When a plane wave falls, the amplitude distribution of the wave field amplitude A_0_ on the crystal surface is constant and can be equal to 1 (unity). The difference in the phase of the plane wave illuminating the crystal surface at different points is already taken into account in the equations of the dynamic theory. In all other cases, it is necessary to consider the deviation of the incident wavefront from a plane wavefront. If, as before, a real radiation source is a set of point sources emitting independently of one another (incoherently), then the wave field of each of these is described by the following expression (one spectral component):(14)$$E_{0}\left(r\right)=\sqrt{I\left(\xi ,\eta \right)}\frac{e^{ik_{0}\left(E\right)r}}{r}=A_{0}\left(r\right)\exp\left(ik_{0}r\right),$$
where:(15)$$A_{0}\left(r\right)=\frac{\sqrt{I\left(\xi ,\eta \right)}}{r}\exp\left[i\left(k_{0}r-k_{0}r\right)\right],$$
which is the required distribution of the wave field amplitude on the crystal surface. The phase factor in Expression (15) describes the deviation of a spherical wavefront from a plane wavefront.

Next to consider is the formation of the diffraction pattern, which is obtained when the wave falls $E_{0}\left(r\right)=E_{0}\left(x,z\right)$ that has passed through the limiting slit to the surface z=0 plane- parallel crystal with thickness *t*, located in Bragg’s geometry. The *x*-axis is directed along the crystal surface, and, in the case of Bragg’s geometry, the output surface of the crystal coincides with the input surface. It is necessary to find the distribution of the field’s $E_{g}\left(x\right)$ diffracted (g=h,z=0) and passed (g=0,z=t) beams.

The field’s $E_{0}$ and $E_{h}$ on the crystal surface z=0 can be represented as:(16)$$E_{g}\left(x\right)=A_{g}\left(x\right)\exp\left(ik_{gx}x\right),$$
where (g=0,h) with:(17)$$k_{0x}=k\cos\left(\theta_{Br}+\psi \right)-s,\;k_{hx}=k\cos\left(\theta_{Br}-\psi \right)-s$$
where the value $s=k\gamma_{0}\Delta \theta$ is determined by the angular alignment of the crystal $\Delta \theta =\theta -\theta_{Br}$, and θ is the angle between incidence k and the system of the reflecting planes of the crystal. The other designations—$\gamma_{0}=\cos\left(k,n\right)=\sin\left(\theta_{Br}+\psi \right),\;\gamma_{h}=-\cos\left(k+h,n\right)=\sin\left(\theta_{Br}-\psi \right)>0$, and ψ—refer to the angle between the reflecting planes and the crystal surface $\left(\left|\psi \right|<\theta_{Br}\right)$.

To find the coordinate dependence of the diffracted wave amplitude, $A_{h}\left(x\right)$ can be decomposed into $A_{0}\left(x\right)$ and $A_{h}\left(x\right)$ on the plane waves:(18)$$A_{g}\left(x\right)=\overset{\infty }{\underset{-\infty }{\int }}A_{g}\left(q\right)exp\left(iqx\right)dq$$
(19)$$A_{g}\left(x\right)=\frac{1}{2\pi }\int_{-\infty }^{\infty }A_{g}\left(x\right)exp\left(-iqx\right)dx.$$

Moreover, from the dynamic theory of X-ray diffraction for the Fourier component (plane waves):(20)$$A_{h}\left(q\right)=r\left(s-q\right)A_{0}\left(q\right),$$
where r(s−q) is the plane wave amplitude reflection coefficient $A_{0}\left(q\right)$ incident at an angle of $\Delta \theta \prime =\Delta \theta -\left(q/k\gamma_{0}\right)$ in relation to the exact Bragg angle [17,18]. In the case of a thick crystal, it is equal to the known expression:(21)$$r\left(\Delta \theta \right)={\left(b\chi_{h}/\chi_{-h}\right)}^{1/2}\left[-y\pm \sqrt{y^{2}-1}\right]$$
where *y* is the normalized angular alignment, expressed in units of the half-width of the diffraction reflection curve; $b=\gamma_{0}/\gamma_{h}=\sin\theta_{0}/\sin\theta_{h}$ is the reflection asymmetry coefficient; and $\chi_{h}\;\mathrm{and}\;\chi_{-h}$ are the Fourier components of the crystal polarizability [10,11].

It is also possible to obtain a form of recording equivalent to Expression (20), which combines not only the Fourier components q of the fields, but also their spatial distributions. To do this, Expression (20) is substituted into (18), and (19) is used for $A_{0}\left(q\right)$, and the variables x−x′=ξ, s−q→q are replaced:(22)$$A_{h}\left(x\right)=\frac{1}{2\pi }\overset{\infty }{\underset{-\infty }{\int }}dq\overset{\infty }{\underset{-\infty }{\int }}r\left(q\right)A_{0}\left(x-\xi \right)e^{i\left(s-q\right)\xi }d\xi .$$

The change in the integration order leads to the following integral relation for the spatial distribution of the diffracted wave $A_{h}\left(x\right)$ at the arbitrary distribution of the incident wave $A_{0}\left(x\right)$:(23)$$A_{h}\left(x\right)=\frac{1}{2\pi }\int_{-\infty }^{\infty }G_{h}\left(\xi \right)A_{0}\left(x-\xi \right)e^{is\xi }d\xi ,\;$$
(24)$$G_{h}\left(\xi \right)=\frac{1}{2\pi }\int_{-\infty }^{\infty }r\left(q\right)e^{-iq\xi }d\xi ,$$
where $G_{h}\left(\xi \right)$ is the Green function of the space-uniform Bragg diffraction problem.

The Green function is, by definition, a function of point source influence. In this case, a point source is located on the crystal surface and determines the degree of blurring of this point in a spot of finite size. Note that the angular spectrum of the point source is uniform in 4*π* steradians or 2*π* radians in the 2D case and linear source. The part of this spectrum that participates in image formation is determined by the crystal diffraction reflection coefficient, which is the Fourier image of the Green function (24). Thus, the Green function describes the blurring of the point as a result of the limited perception of the input angular spectrum by the crystal analyzer.

Thus, as a result of theoretical consideration, a formalism is obtained that allows the influence of the crystal analyzer in the optical scheme to be taken into account if the spatial distribution of the wave field on the crystal surface (23) or the angular spectrum of incident radiation (20) is known. Similar expressions can be obtained using a crystal analyzer in Laue geometry.

This section, with a simple numerical example, can be completed to present the influence of the crystal analyzer at a qualitative level. Let a spherical wavefront fall on the crystal with a slit located at a certain distance in front of it. Figure 2 shows the modulation of the beam cross-section in the system of the slit analyzer at different normalized angles of adjustment of the crystal analyzer. That is, angular alignment of the crystal analyzer leads to changes in the spectrum of the spatial frequencies of the object: rather than homogeneous illumination after the slit (Figure 2b), there is a transverse modulation of the beam intensity, depending on the value of angular alignment (Figure 2a,c). Calculations are made for a synchrotron-type radiation source located at a distance of 40 m. Nevertheless, even in this case, the crystal analyzer can feel the wavefront curvature.

Figure 3 demonstrates the causes of transverse modulation of the beam. Cases (a) and (c) in Figure 2 correspond to the choice of the working point (angle alignment) of the analyzer on the front and back slopes of the rocking curve, respectively. Due the diffraction on a slit, in a falling beam, there is a spread in wave vectors, i.e., in the angles of incidence on the crystal surface. In this case, depending on the choice of the working point (angle alignment) of the crystal, a part of the radiation falls on the peak of the rocking curve under the exact Bragg conditions, and another part is out Bragg diffraction conditions, which leads to transverse modulation in the beam section.

The ability to convert the phase modulation of diffracted radiation (angular spectrum) into visible intensity distribution through the diffraction reflection from the perfect crystal is the basis of X-ray phase-contrast methods in monochromatic radiation in a scheme with the crystal analyzer [9,19,20]. In contrast, only the phase contrast can achieve submicron resolution in the hard X-ray radiation.

When using polychromatic laboratory X-ray sources (Figure 4), interpretation of the obtained projection images becomes even more complicated. At present, no complete description exists of the algorithm for solving the considered problem. Only the first experimental results have been obtained, which are presented in the next section.

## 4. Tomographic Projection Post Processing

### 4.1. Experimental Results

Recently [8], we proposed a method of alignment of a crystal analyzer to select two characteristic lines from the spectrum of a conventional X-ray tube for simultaneous registration of tomographic projections. When using a crystal analyzer, projection images at different wavelengths are separated in space and can be registered independently for further processing, which is of interest for the spectral tomography method. Experiments were carried out on the X-ray diffractometer [8], the scheme of which is presented in Figure 5.

A silicon crystal analyzer with a thickness of 540 µm was mounted on a goniometer perpendicular to the X-ray beam (Figure 5). The source of X-ray radiation was an X-ray tube with a molybdenum anode with a focus size of 0.4 × 12 mm. Taking into account the size of the focal spot of the source, which was ~1 mm, the size of the probe crystal of the polychromatic X-ray beam was regulated by two mutually perpendicular slits and was 2 mm both vertically and horizontally. Initially, the crystal was set so that its (111) plane was perpendicular to the beam. By rotation around the X, Y, and Z axes (Figure 5), the analyzer was adjusted to the maximum reflections in Laue geometry for the crystallographic plane $\left(\bar{1}11\right)$ in the case of the $K_{\alpha }$ line and $\left(1\bar{1}1\right)$ for the $K_{\beta }$ line. The crystal analyzer alignment is described in more detail in [8]. The X-ray images of two characteristic lines transmitted through a 5 mm thick aluminum filter beam were recorded on a 2D charge-coupled device CCD camera (Ximea XiRay11). The exposure time of one frame was 10 s. The size of the sensitive element (pixel size) of the detector was 9 µm. The source–crystal distance was 1000 mm and the crystal–detector distance was 22 mm.

The results obtained by the example of a test object (calibration grid) are shown in Figure 6a. In the image of the test object (Figure 6a), on the left, there is an absorption contrast of $K_{\beta }$ on reflection from the peak of the rocking curve; on the right, there is a quasi-phase-contrast in $K_{\alpha }$ on reflection with angular adjustment. The image on the right appears in greater relief as a stereo image, which is typical for the phase-contrast registration mode.

### 4.2. Data Processing Algorithm

To interpret the results of the experiments, the expressions that link the values registered by the detector pixel to the parameters of the object are written and the crystal analyzer is placed in the optical path. Let us start with the image registered in the transmission mode and rewrite Expression (10), taking into account the set position of the crystal analyzer:(25)$$I_{T}\left(x_{\det},y_{\det}\right)=\int^{\;}dEI_{0}\left(x_{\det},y_{\det},E\right)exp\left(-\int_{0}^{L\left(cx_{\det},y_{\det}\right)}\mu \left(z;E\right)dz\right)\left(1-F_{refl}\left(E_{i},d_{cr},\varphi ,\;\vartheta \;\right)\right)$$
where $\left(x_{\det},y_{\det}\right)$ is the intensity recorded by the detector cell, the position of which is set in a pair, i.e., $x_{\det}\;\mathrm{and}\;y_{\det}$; *L* is a line passing through the source (assuming that it is a point) and a point specifying the position of the detector cell; and φ and ϑ are the azimuthal and tangential angles that determine the orientation of the family of reflecting planes relative to the incident beam, respectively. The difference between Expressions (25) and (10) is that an operator describing the loss of the transmitted intensity due to the deviation of the X-ray assembly part from the direction of probing appears. Approximation of the intensity attenuation due to the deviation of a part of the rays, i.e., the type of function $F_{refl}\left(E_{i},d_{cr},\varphi ,\;\vartheta \;\right)$, remains a challenge that needs to be addressed. The quasi-monochromatic images $I_{k\alpha }$ and $I_{k\beta }$ formed by a crystal are projectively distorted. To compensate for the distortions that result from asymmetric diffraction of the selected families of reflecting planes, we used the projective transformation H:(26)$$HI_{k\alpha }^{i}=I_{T}^{i},\;i=\bar{1,4},$$
where *i* is the number of projective base points. The transformation is performed over the coordinates of the projective basis points [22,23,24,25]. The lattice used as a calibration object does not limit the class of test objects because the theory of constructing projective-invariant bases for projective transformed smooth convex figures is also developed [26].

When working with a test object, three images—transmission and two images in the $K_{\alpha }$ and $K_{\beta }$ lines—are used. When recording a transmission image, the attenuator (a 5 mm thick aluminum filter) is placed in front of the detector screen in a straight beam area, i.e., the signal conditioning model is complemented by the filter attenuation:(27)$$I_{T}\left(x_{\det},y_{\det}\right)=\int^{\;}dEI_{0}\left(x_{\det},y_{\det},E\right)exp\left(-\int_{0}^{L\left(x_{\det},y_{\det}\right)}\mu \left(z;E\right)dz\right)F_{abs}\left(E,d_{cr}\right)\left(1-F_{refl}\left(E_{i},d_{cr},\right)\right)exp\left(-\mu^{Al}d^{Al}\right).$$

An expression that links the measured intensity without a sample to the crystal and filter parameters is:(28)$$I_{T}^{0}\left(x_{\det},y_{\det}\right)=\int^{\;}dEI_{0}\left(x_{\det},y_{\det},E\right)\left(1-F_{refl}\left(E_{i},d_{cr},\right)\right)exp\left(-\mu^{Al}d^{Al}\right).$$

By dividing (28) by (27) and taking the logarithm, we can approximate the data linearization procedure, i.e., the ratio commonly used as a fast preprocessing step for tomographic data before reconstruction:(29)$$\int_{0}^{L\left(x_{\det},y_{\det}\right)}\bar{\mu }\left(x_{\det},y_{\det},z\right)dz=ln\left(\frac{I_{T}^{0}\left(x_{\det},y_{\det}\right)}{I_{T}\left(x_{\det},y_{\det}\right)}\right)$$

Note that the sub-integral function $\bar{\mu }\left(x_{\det},y_{\det},z\right)$ describes the distribution of an averaged source spectrum attenuation coefficient. The absolute value of the coefficient does not allow the composition to be estimated. The spectrum is weakened by the object and changed by the crystal analyzer installed in the path. When performing a division operation, the contribution is not scaled.

Figure 7 shows the normalized measurement results of the calibration grid $P_{*}^{norm}$. To calculate the normalized value:(30)$$P_{*}^{norm}\left(x_{\det},y_{\det}\right)=\frac{I_{*}^{0}\left(x_{\det},y_{\det}\right)-I^{dark}\left(x_{\det},y_{\det}\right)}{I_{*}\left(x_{\det},y_{\det}\right)-I^{dark}\left(x_{\det},y_{\det}\right)}.$$
the dark current $I^{dark}\left(x_{\det},y_{\det}\right)$ is measured. Subscript “*” takes on one of three meanings: T (transmission) (Figure 7a), $K_{\alpha }$ (Figure 7b), or $K_{\beta }\;$(Figure 7c).

Images in the reflections ($K_{\alpha }$ and $K_{\beta })$ differ from transmission signals in the manner they are formed. These are pseudo-monochromatic images, meaning that the attenuation coefficient or the integrated function μ after tomographic reconstruction estimates the linear attenuation coefficient of the object, which can already be associated with the chemical composition of the object.

Figure 7b,c shows the normalized images formed by the crystal $P_{K_{\alpha }}^{norm}$ and $P_{K_{\beta }}^{norm}$, respectively. Figure 7d,e shows the results of the projective correction of the images $P_{K_{\alpha }}^{proj}$ (Figure 7b) and $P_{K_{\beta }}^{proj}$ (Figure 7c), respectively.

Now it is possible to align images pixel-to-pixel with the same coordinates. To visualize the results of the alignment, we constructed a three-channel color image, which is shown in Figure 8. The following algorithm was used for color image calculation. In the first step, we built a mask to minimize the contribution of the pixels that did not contain the object and pixels of the object with high noise dispersion. The following steps were performed for the pixels of the mask: narrowing of the dynamic range of pixel values to increase the contrast to the range of 0.3–0.95; channel auto-calibration of the images to equalize the brightness range of the monochrome channels; and a three-channel image was built using the linear method. The red channel contained transmission image values, the green channel contained pixels of $P_{K_{\alpha }}^{proj},$ and the blue channel contained pixels of $P_{K_{\beta }}^{proj}$. To demonstrate the capabilities of the method, we present the measurement results of a chip section in a circuit with a crystal analyzer. We collected 50 images, each with 3 s exposure, in the transmission mode and 50 images, each with 30 s exposure, in the two reflexes. The averaged and correlated images, according to the procedure described above, are shown in Figure 9.

After applying the projective transformation to the pixels of each of the $K_{\alpha }$ and $K_{\beta }$ line images, we constructed a three-channel image, which is shown in Figure 10.

From this image, it can be concluded that the proposed procedure to compensate for the geometric distortions caused by the crystal analyzer installed in the optical path is correct. Images of each of the channels may serve as fully fledged tomographic projections, being shot at different angles of the object rotation. The use of this measurement scheme allows the object to be placed on the goniometer, i.e., to implement the rotation of the sample.

Correction of geometric distortions allows a complete set of polychromatic projections to be created. The presence of a crystal analyzer in the optical path complicates the model of projection formation, but the development of methods for solving the problem of tomographic reconstruction, taking into account the changing phase of the probing radiation, is ongoing [27,28,29]. The color image is not used in the reconstruction procedure. Before the optimization problem solution, each type of collected projection is preprocessed separately. The transmission projections are preprocessed according to Expression (27), and the spectral projections are preprocessed according to the model described in Section 3.

## 5. Conclusions

This article demonstrated how a combination of well-known tools—i.e., a standard 2D detector (CCD camera) and a crystal analyzer—can improve tomographic sensing multimodality. The use of a crystal analyzer made it possible to isolate two characteristic lines of the molybdenum anode from the polychromatic radiation of a standard X-ray tube, namely, $K_{\alpha }$ and $K_{\beta }$. Three radiographic projections (images) were recorded as a result of the experiment. The projection images at different wavelengths were separated in space and can be registered independently for further processing, which is of interest for the spectral tomography method. The first experimental results presented here appear promising.

In the current work, an algorithm for the geometric correction of projection distortions was derived. Furthermore, work began on building a model of tomographic projections based on the dynamic theory of X-ray diffraction in crystals, which will allow quantitative analysis. The fact that the incident beam is polychromatic greatly complicates the task, but is nonetheless of significant interest. The extension of multimodality to the ability to switch between detection modes—i.e., absorbing contrast or phase-contrast imaging—will clearly reward the expended effort.

It also appears promising to use such a spectral pattern with a crystal analyzer in Bragg’s geometry. In this case, it is possible not only to achieve a gain in the intensity of diffraction monochromatic projections, but also to avoid the problem of deformation of the crystal due to its thickness, which is present in Laue geometry. In addition, Bragg’s geometry is simpler in terms of crystal alignment and quantitative analysis of the results, because there are no thick oscillations on the thick crystal rocking curve. However, its implementation requires a significant upgrade of experimental hardware.

Unfortunately, due to the lack of a suitable test object, it was not possible to demonstrate another multimodality extension of the proposed detection scheme. Namely, when a test object includes elements with an absorption edge lying within the range of $K_{\alpha }$ and $K_{\beta }$ of the used anode, a map of the distribution of this element should be obtained in (diffraction) monochromatic projections. Additional expansion in this direction can be achieved by varying the choice of the material of the used anode. These are tasks to be pursued in the near future. In addition, the use of a crystal analyzer as an energy separator of the primary beam together with polychromatic synchrotron X-ray radiation opens new opportunities for research to contrast almost any substance in the sample. In this case, the crystal can be tuned to obtain reflections before and after the absorption edge of a particular element.

## Figures and Tables

**Figure 1 sensors-20-06970-f001:**
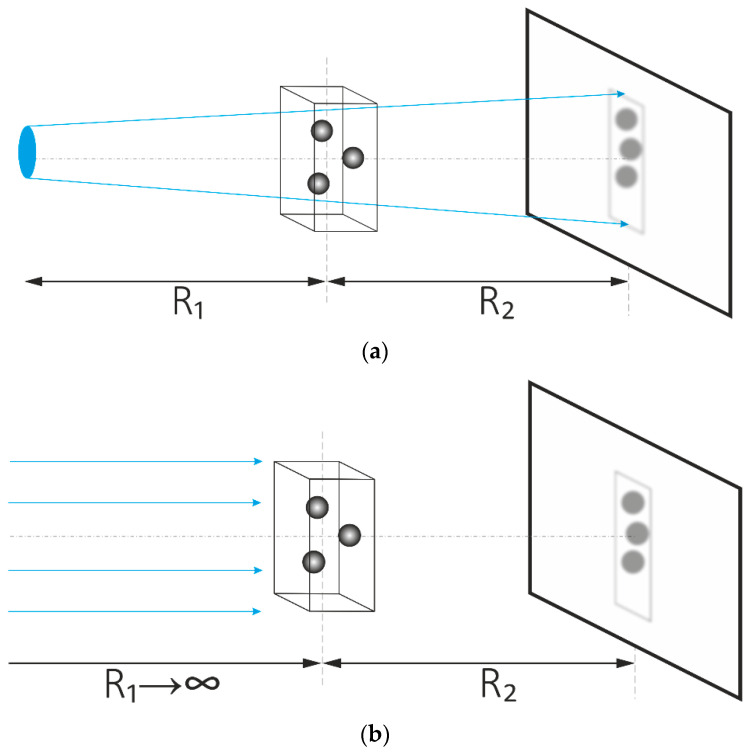
The formation of tomographic projections from physical to mathematical models: (**a**) standard scheme of tomography; (**b**) approximation of a plane wave beam; (**c**) approximation of the absence of diffraction in free space; (**d**) 3D object reduced to a 2D absorbing and phase shifting screen.

**Figure 2 sensors-20-06970-f002:**
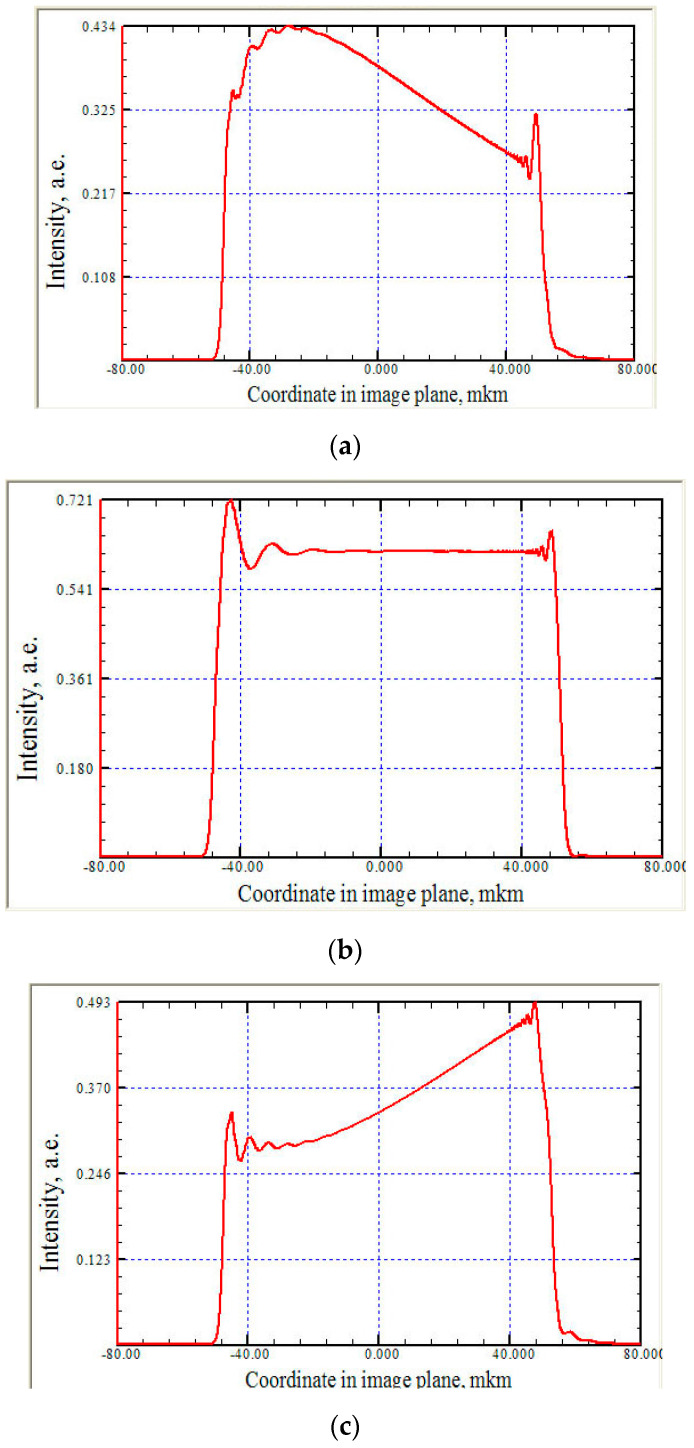
Modulation of the beam cross-section in a slit monochromator system at various normalized angles of alignment of the crystal analyzer (reflection Si (220), Cu $K_{\alpha }$, *θ*_Br_ = 23.652°): (**a**) y = 1.05; (**b**) y = 0; (**c**) y = -1.05. Other parameters: An extended rectangular source *ξ*_s_ = 100 μm, located at a distance of 40 m, illuminates a slit size of a = 100 μm; the distance of the slit analyzer is 5 cm; the asymmetry coefficient of the crystal b = 1.

**Figure 3 sensors-20-06970-f003:**
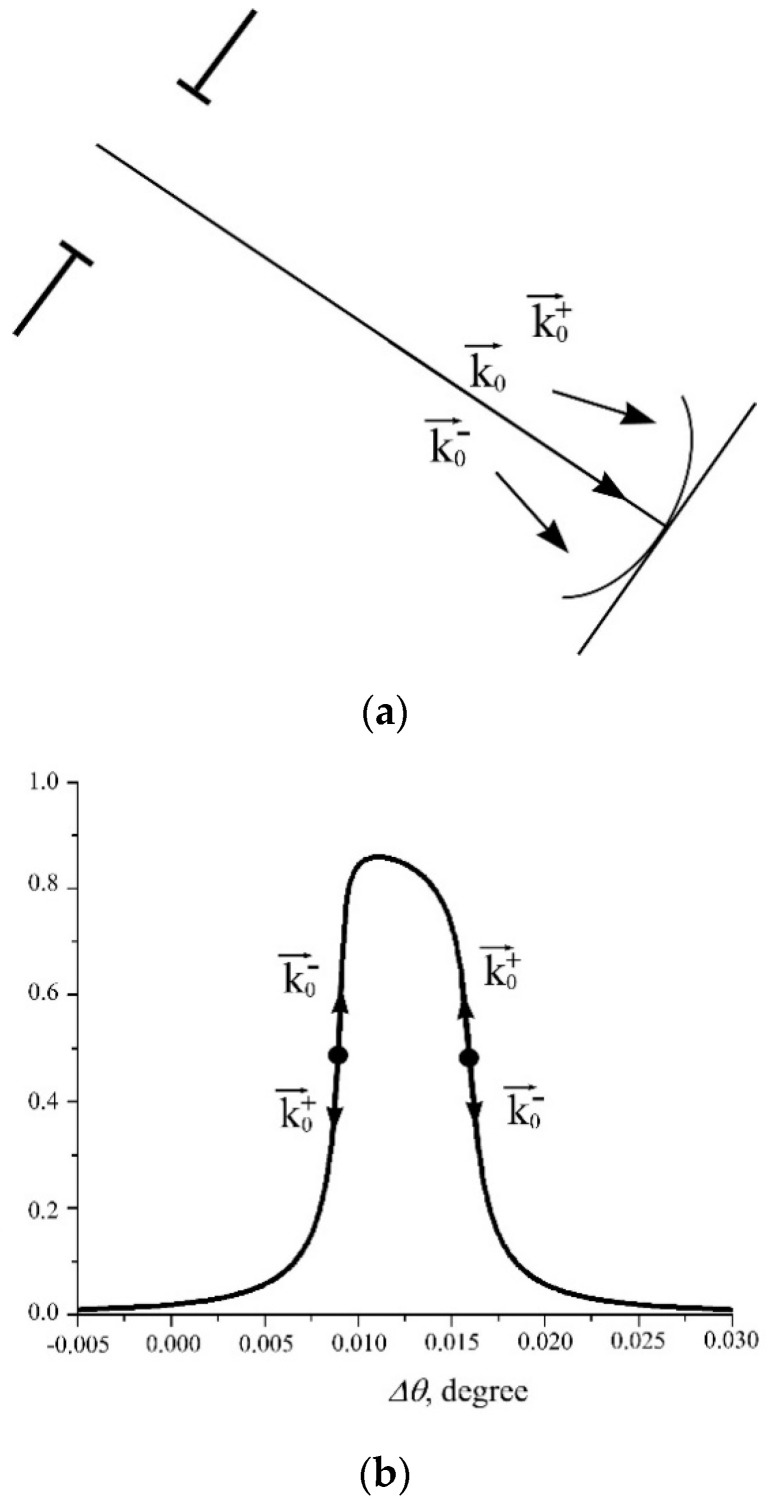
Interpretation of transverse beam modulation in the reflection from the crystal analyzer: (**a**) decomposition into wave vectors (plane fronts); (**b**) contrast formation on the curve of the diffraction reflection.

**Figure 4 sensors-20-06970-f004:**
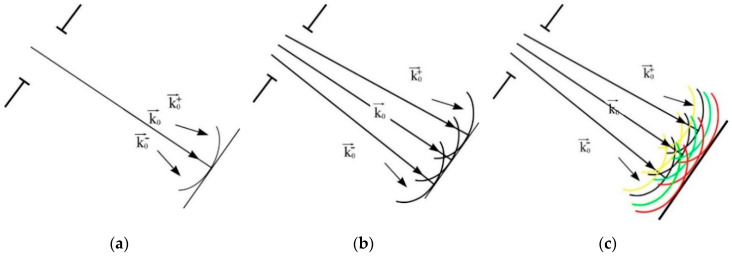
Decomposition into wave vectors (plain fronts) in the case of: (**a**) a point source; (**b**) an extended source; and (**c**) an extended polychromatic source.

**Figure 5 sensors-20-06970-f005:**
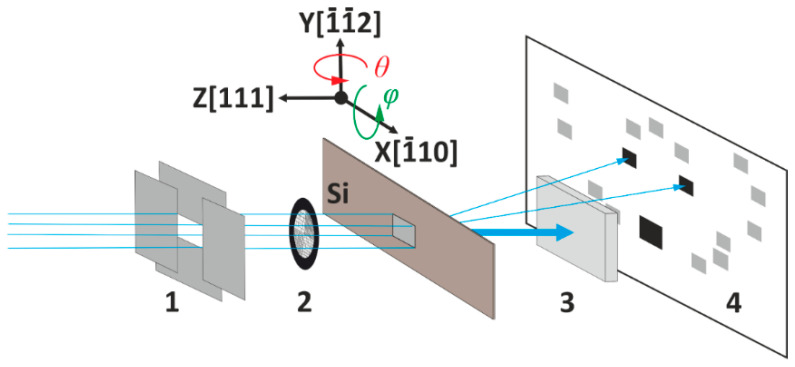
The scheme of the experiment: 1—system of slits; 2—sample in the study (grid); (3—aluminum filter; 4—charge-coupled device (CCD) camera.

**Figure 6 sensors-20-06970-f006:**
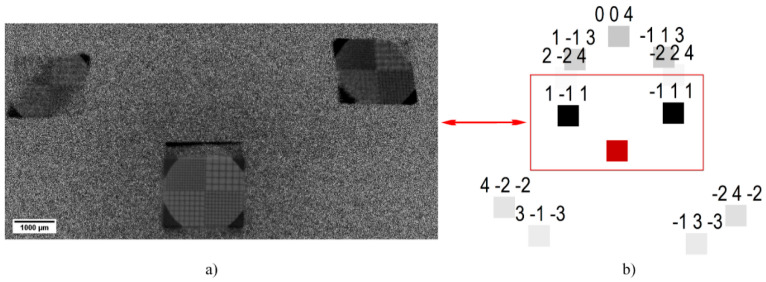
Image-type reflections: (**a**) with calibration grid for cases of the transmitted X-ray beam, the planes $\left(\bar{1}11\right)$ and $\left(1\bar{1}1\right)$ correspond to the characteristic $K_{\alpha }$ and $K_{\beta }$ lines of Mo accordingly; (**b**) theoretical simulation using LauePt software [21].

**Figure 7 sensors-20-06970-f007:**
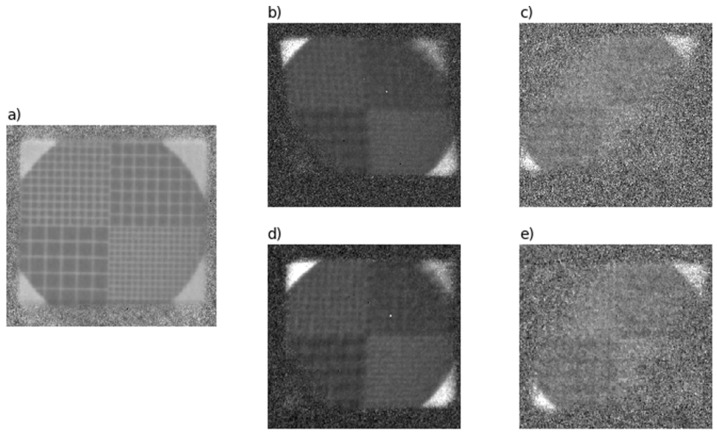
(**a**) Image in transmission mode; (**b**) $K_{\alpha }$ image $P_{K_{\alpha }}^{norm}$; (**c**) $K_{\beta }$ image $P_{K_{\beta }}^{norm}$; (**d**) $K_{\alpha }$ image after projective correction $P_{K_{\alpha }}^{proj}$; (**e**) $K_{\beta }$ image after projective correction $P_{K_{\beta }}^{proj}$.

**Figure 8 sensors-20-06970-f008:**
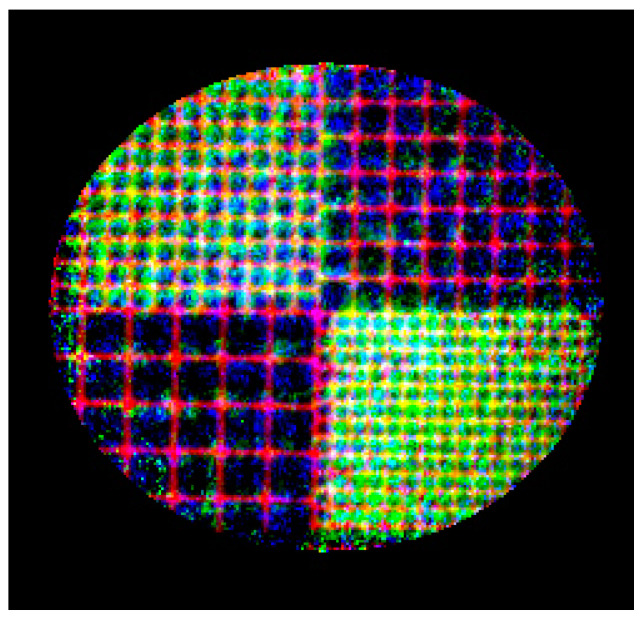
A three-channel image of the calibration grid. The red channel contains $P_{T}^{norm}$, the green channel contains $P_{K_{\beta }}^{proj}$, and the blue channel contains $P_{K_{\alpha }}^{proj}$.

**Figure 9 sensors-20-06970-f009:**
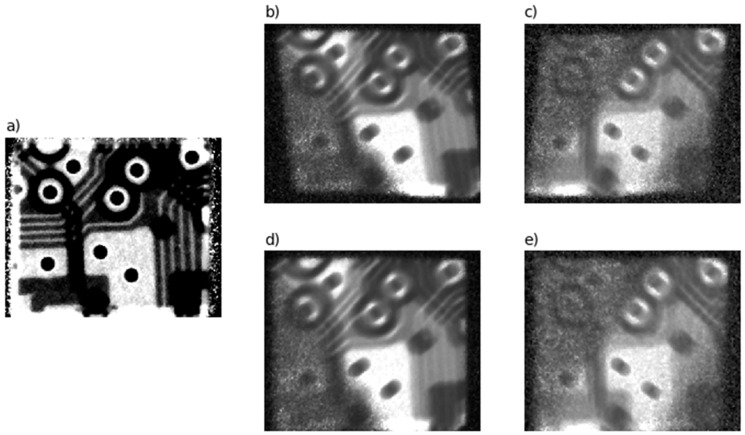
(**a**) Normalized images of the chip section; (**b**)$\;P_{K_{\alpha }}^{norm}$ image; (**c**) $P_{K_{\beta }}^{norm}$ picture; (**d**) after projective correction $P_{K_{\alpha }}^{proj}$; (**e**) after projective correction $P_{K_{\beta }}^{proj}$.

**Figure 10 sensors-20-06970-f010:**
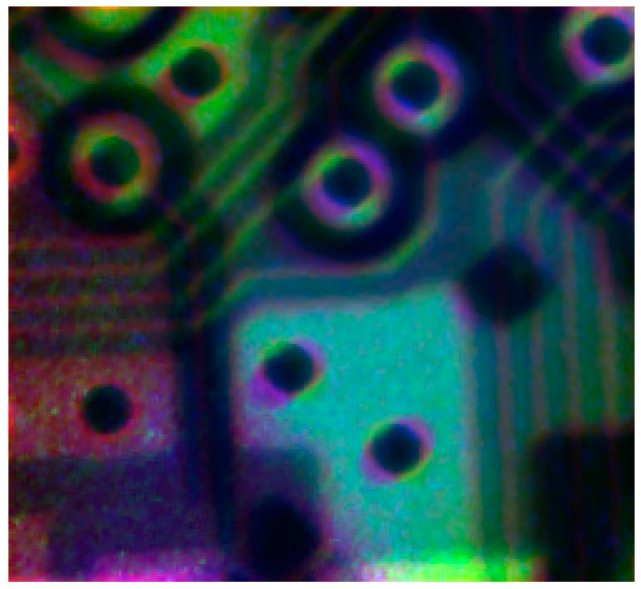
Color image of the chip section. The red channel contains $P_{T}^{norm}$, the green channel contains $P_{K_{\beta }}^{proj}$, and the blue channel contains $P_{K_{\alpha }}^{proj}$.

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
