# Peer review of "Enhanced Tomographic Sensing Multimodality with a Crystal Analyzer"

_sensors, 2020, doi:10.3390/s20236970_

Round 1
Reviewer 1 Report
Report on ms. sensors-968663
“Enhanced tomographic sensing multimodality with crystal-analyzer” by Buzmakov et al.
The manuscript explores the potential offered by a crystal analyzer and Bragg diffraction to add to a usual polychromatic attenuation radiograph, the diffraction image of the K_\alpha or K_\beta wavelength of the source. The idea is interesting and although it was first introduced in Ref. 8, the concept is further explored and an experimental example is shown.
However, manifestly the paper has been written hastily and presumably not read for corrections. The wording is very unconventional, especially in the more technical section. Many sentences lack a verb. Typography is random. Many typos are present including in figures (Fig. 1b for instance is identical to 1c) or in equations (for example Eq. 17 lhs should be $A_h(r)$).
If one tries to go beyond this superficial level, despite the interest of the problem setting, the paper should be simplified. The problem is introduced in its full complexity before being simplified, and as a result, it is difficult to apprehend what kind of approximations are involved and how severe they are.
Although the tile evokes tomography, a single orientation is considered, for which a direct attenuation image is obtained together with two diffraction images. This is by itself of interest, but obviously the proposed presentation oversells the results. For instance, the proposed approach requires a 2D image (as shown in fig. 1d), and hence the projection of the diffraction image does not consider the thickness and as a consequence, the holes in the chip section become elongated in the two diffraction images (see fig. 9) and hence registration of the three images onto each other miss completely this thickness effect, and there is no chance that the proposed method may one day reconstruct a spectral tomography. from the registration of the three images over a complete revolution of the scanned sample.
There are indeed ways to enrich a tomography with this technique, but not from the rough approximation that is proposed here. Considering that the initial paper by one of the authors was just published, I suggest that the authors do not rush to publish too loosy or unusable results, but rather take their time to polish their methodology.
In its present state, I suggest rejection of the manuscript.
Author Response
Please, see the attachment.

Reviewer 2 Report
The paper discusses how use of a standard CCD-camera in combination with a crystal analyzer can improve tomographic sensing multimodality in X-ray imaging. By using the molybdenum anode lines to produce simultaneous projections of the target, information for spectral tomography can be extracted.
In general the paper is well written and it provides detailed information about the the theoretical background, and about the post-processing that is required for tomographic reconstruction. The reader will have a good view about the benefits of using the discussed method.
However, it seems that the manuscript has not been proofread properly. To avoid the problems with the language, you could simplify the sentences and remove the 'chatty' part of the text.
Reviewer 3 Report
The authors discuss an improvement for x-ray imaging by installing a Laue-analyzer between object and detector, which reflects both the Kα and Kβ fluorescence lines into different parts of the area detector. This results in three different images of the object on the detector, which can be combined into one image with some mathematical transformations. This combined image has more information than a regular transmission image. The authors suggest that this method can be applied for contrast-imaging methods, if the anode of the x-ray tube is selected such that the energy of the absorption edge of one of the critical elements in the sample is in between the energies of the Kα and Kβ fluorescence lines of the anode.
I recommend to publish the manuscript as a regular article in sensors, however, several modifications are required.
The sentence on page 2, lines 4-5 ‘X-ray imaging is a logical continuation of the X-ray imaging technique, allowing ...’ is not complete. The first ‘X-ray imaging’ requires additional description, like ‘X-ray imaging at different energies’ or ‘X-ray imaging at different sample-angles’.
The last sentence of the first paragraph of the conclusions on page 18 might require also some work:'The first experimental results presented here look promising and promising.'
On page 4, the authors give examples for I(ξ) and g(E) for synchrotron radiation sources. The equation for I(ξ) is more or less correct, however, the equation for the energy distribution does not describe any synchrotron radiation source. Bending magnets and Wigglers have a continuous energy spectrum from the IR to the hard x-ray range (>100keV), and undulators emit rather narrow bands with a width of less than 2keV. The use of monochromators or multilayers can reduce the bandwidth, however, the width is rather narrow (1-5eV for a monochromator, 20-100eV for a multilayer).
Page 13, first line: I doubt that the Si crystal analyzer had a thickness of 520mm. 520 μm is more reasonable.
Figure 7: It might be helpful to mention that the (-1 1 1) reflection diffracts the Kα lines, and the (1 -1 1) reflection the Kβ line.
Figure 8a: Please describe the different color channels in the figure caption (similar to the text).
Figure 9 shows that some of the wires become better visible in the reconstructed image of the transmitted and diffracted pictures, however, some of the structures, especially the 'holes' in the middle of the lower half become smeared in the direction of diffraction. A more detailed discussion of those two effects would be useful. Is the distortion an artifact of the projective correction, or is it related to the thickness of the chip? I also don’t see much of a difference between figures 9 b) and d), and c) and e) respectively. The distortion seems to be smaller in these images than that in figure 7 of the reference sample.
In my humble opinion, the application of Bragg geometry is not so trivial. It is easier to align the main reflection, however, one diffracts just one energy, and the crystal has to be rotated in order to get also a diffraction pattern of the other fluorescence line. Additionally, the transmission signal will be blocked since crystals for Bragg-geometry are thicker (for stiffness) and the Bragg-angle at a lower angle increases the effective thickness. It is not possible to detect all three images simultaneously.
I see a rather limited application of the contrast method for imaging, which the authors describe in the introduction. Yes, for every element heavier than Ti, there is another element (usually Z+1) which has Kα and Kβ fluorescence lines, which have an energy above and below the energy of the K-absorption edge, however, it is not trivial to use every element as an anode in an X-ray tube. The authors mention themselves that they did not have an appropriate test-structure available, which contained either Zr or Nb. I agree that the contrast method will give better results, but it will remain a niche-application.
